# Comparison of the Degradation Performance of Seven Different Choline Chloride-Based DES Systems on Alkaline Lignin

**DOI:** 10.3390/polym14235100

**Published:** 2022-11-24

**Authors:** Penghui Li, Yuan Lu, Xiaoyu Li, Jianpeng Ren, Zhengwei Jiang, Bo Jiang, Wenjuan Wu

**Affiliations:** 1Jiangsu Co-Innovation Center of Efficient Processing and Utilization of Forest Resources, Nanjing Forestry University, Nanjing 210037, China; 2College of Light Industry and Food Engineering, Nanjing Forestry University, Nanjing 210037, China

**Keywords:** alkaline lignin, choline chloride, deep eutectic solvent (DES), degradation

## Abstract

Lignin is a natural polymer second only to cellulose in natural reserves, whose structure is an aromatic macromolecule composed of benzene propane monomers connected by chemical bonds such as carbon–carbon bonds and ether bonds. Degradation is one of the ways to achieve the high-value conversion of lignin, among which the heating degradation of lignin by deep eutectic solvent (DES) can be an excellent green degradation method. In this study, choline chloride (CC) was used as the hydrogen bond acceptor, and urea (UR), ethylene glycol (GC), glycerol (GE), acetic acid (AA), formic and acetic mixed acid (MA), oxalic acid (OX), and *p*-toluenesulfonic acid (TA) were used as hydrogen bond donors to degrade lignin. NMR hydrogen spectroscopy was used for the simple and rapid determination of phenolic hydroxyl groups in lignin. FT-IR spectroscopy was used to characterize the changes of functional groups of lignin during DES treatment. GPC observed the molecular weight of lignin after degradation and found a significant increase in the homogeneity (1.6–2.0) and a significant decrease in the molecular weight M_w_ (2478–4330) of the regenerated lignin. It was found that acidic DES was more effective in depolymerizing alkaline lignin, especially for the toluene–choline chloride. Seven DES solutions were recovered, and it was found that the recovery of DES still reached more than 80% at the first recovery.

## 1. Introduction

With the increasing depletion of non-renewable energy sources worldwide, the conversion of biomass to fuels and chemicals has taken the stage due to its sustainability [1]. Currently, China is a country where agriculture occupies the majority of the country and a large number of agricultural wastes are directly discarded or polluted and burned, such as some walnut shells and rice husks. If these agricultural wastes are utilized, not only can environmental problems be avoided, but also agricultural economic growth will be stimulated. The most abundant and not easily decomposed agricultural waste is lignin [2]. Lignin accounts for 20–30 wt% of lignocellulosic biomass, and its abundance in nature is still very considerable [3]. In addition to being obtained directly from nature, a large amount of lignin from the paper industry is available in the black liquor formed by the kraft and sulfite pulping processes that are in urgent need of industrial treatment. Industrial lignin is poorly utilized (less than 2%), and the remaining lignin is only discarded or used as a low-value fuel [4]. From the structural point of view, lignin is a complex aromatic polymer derived from the corresponding *p*-hydroxycinnamyl alcohol, consisting of pineol, mustard alcohol, and *p*-coumaryl alcohol derivatives forming the benzene propane unit. The presence of aromatic rings, methoxy, carboxyl, and hydroxyl groups in lignin makes it chemically active [5]. However, the structure of lignin is relatively recalcitrant and complex, making it difficult to upgrade [6]. The bonds in lignin are mainly ether bonds (β-*O*-4′, α-*O*-4′, 4-*O*-5′) and carbon–carbon bonds (β-β′, β-1′, 5-5′, α-1′, β-5′), where the β-*O*-4′ bond is the main bond type of lignin, accounting for a large percentage, and is more easily cleaved, so it is preferred for various depolymerization reactions [7]. Due to the complex structure of lignin macromolecules, the conditions of lignin depolymerization reactions are often harsh and the yields are not very high, causing the high quality and high value of lignin to be hindered [8].

Currently, the conversion of lignin to aromatic chemical monomers has become a hot topic. Common lignin depolymerization methods include hydrolysis [9], pyrolysis [10], hydrogenolysis [11,12], and oxidation [13,14]. The conventional depolymerization methods, although strongly reactive, are relatively environmentally unfriendly and non-energy-efficient, involving high reaction temperatures, high pressures, or expensive and toxic catalysts [15]. Deep eutectic solvent (DES) is a new green solvent, consisting of a combination of a hydrogen bond donor (HBD) and a hydrogen bond acceptor (HBA), which can be an inexpensive, efficient, environmentally safe, and favorable solvent environment for lignin dissolution [16,17]. DES has good solubility properties for lignin fractions, where DES composed of lactic acid, malic acid, oxalic acid, and propionic acid as hydrogen bond donors, and betaine, ChCl, and urea as hydrogen bond acceptors, showed a high solubility ability for lignin [18]. What is more, DES can be used not only as a solvent but also as a catalyst for many chemical reactions [19].

It has been found that during DES pretreatment, it can effectively cleave the β-*O*-4′ bond and lignin undergoes depolymerization, which is a landmark step for the value addition of lignin to small molecule monomers [20]. DES has also been extensively reported in the extraction and degradation of lignin. Acid DES systems can depolymerize lignin well; Hong et al. [21] used DES consisting of choline chloride/formic acid (1:2 molar ratio) for the degradation of alkaline lignin, and, not surprisingly, most of the β-*O*-4′ bonds were broken and the carbon–carbon bonds (i.e., β-β′, β-5′ bonds) were partially broken, and many phenolic substances were generated. Tan et al. [22] evaluated the structural and thermal properties of lignin using two DES systems, choline chloride/lactic acid and choline chloride/formic acid. They found that more than half of the phenolic compounds formed after many small lignin monomers, and the β-*O*-4′ bond of lignin in DES cleaves faster in a pure lactic acid system. Li et al. [23] used choline chloride and *p*-toluenesulfonic acid as DES systems and degraded alkaline lignin at mild reaction temperatures with an average molecular weight reduction from 17,680 g/mol to 2792 g/mol; the degradation products were phenols, ketones, and a small amount of aldehydes. Basic DES systems (e.g., choline chloride/urea) usually depolymerize lignin by breaking ether bonds, which eventually leads to depolymerization or repolymerization reactions. Neutral DES systems (e.g., choline chloride/glycerol) only slightly alter the lignin structure and are not as capable of breaking the β-*O*-4′ bond [24]. Yu et al. [25] investigated the effect of choline chloride/methanol as the reaction solvent, supplemented with a catalyst, on the selectivity of catalytic oxidation of alkaline lignin. Under fully oxidized conditions, the aromatic region underwent oxidation to produce acetyl vanillin, along with small amounts of vanillin and vanillic acid. The condensation and depolymerization reactions of the side chain groups of alkaline lignin may occur simultaneously in the choline chloride/methanol system, which is the same mechanism of depolymerization as that of the choline chloride/lactic acid system. Therefore, we conclude that the DES system may have the prospect of acting as a solvent or catalyst for lignin degradation reactions. To test this hypothesis, we describe the degradation of alkaline lignin by acidic, neutral, and basic choline chloride DES, hoping to explore its degradation mechanism.

## 2. Materials and Methods

### 2.1. Materials

The lignin material was purchased from a Nanjing pulp and paper mill (China). The number-average and weight-average molecular weight of the alkaline lignin were 2041 and 7349, respectively, and the polydispersity was 3.601 (determined by a GPC instrument LC-20A, Shimadzu, Kyoto, Japan) with eluent tetrahydrofuran (THF). Choline chloride (98.0~101.0%), urea (≥98%), ethylene glycol (≥99.5%), glycerol (≥98.0%), formic acid (≥99.5%), acetic acid (≥99.5%), oxalic acid (≥99.5%), *p*-toluenesulfonic acid (≥99.0%), and other chemicals were purchased from Sinopharm Chemical Reagent Co., Ltd. (Shanghai, China) without further purification.

### 2.2. Elemental Analysis of Lignin

Elemental analyses of the alkaline lignin samples were performed on an Elementar Vario EL cube analyzer. Table 1 lists the results. The content of C, H, N, and S were measured directly from elemental analysis.

### 2.3. Characterization and Analysis of the Products

Weigh 10 mg of dry lignin product, dissolve it in slightly alkaline water, add FC reagent, make up to volume, add 20% Na_2_CO_3_ after 5 min, and then make up to 50 mL. After the mixture was stirred for 2 h, the absorbance of lignin and regenerated lignin at 760 nm was measured using an ultraviolet spectrophotometer (TU-1900, Beijing, China) to obtain the content of phenolic hydroxyl groups in the lignin samples. The structure and functional groups of the products were examined by Fourier-transform infrared spectroscopy (FT-IR) on a VERTEX 80V FTIR spectrometer (Bruker, Karlsruhe, Germany). The scanning range was 500–4000 cm^−1^, and the scan number was 32. Acetylation of lignin was done before molecular weight testing. The gel permeation chromatography determined the molecular weight and dispersion of lignin samples (LC-20A, Shimadzu, Kyoto, Japan). The concentration of lignin in tetrahydrofuran (THF) was about 5 mg/mL. The column temperature was 40 °C, THF was the eluent, and the flow rate was 1 mL/min. The average molecular weight of lignin was measured by an external standard method, in which monodisperse polystyrene was applied as the standard compound. ^1^H NMR was acquired by dissolving 12 mg of acetylated lignin sample in 0.5 mL of deuterated dimethyl sulfoxide and shaking to dissolve it. The ^1^H spectral range was 0–16 ppm (9615 Hz) with 2048 sampling points in the ^1^H dimension and a relaxation time of 1.5 s, accumulated 64 times. The hydrocarbon coupling constant is 145 Hz, and the number of sampling points in the ^1^H dimension is 2048, with a relaxation time of 1.5 s, totaling 64 times.

### 2.4. Lignin Depolymerization Process and DES Recycling

The degradation of lignin in choline chloride-based DES was carried out in the steps shown in Figure 1. First, alkaline lignin was dissolved in a DES solution with a 5 wt% lignin solution ratio (choline chloride as the hydrogen bond acceptor and different hydrogen bond donors as variables). The hydrogen bond donors are *p*-toluenesulfonic acid, oxalic acid, acetic acid, urea, glycerol, ethylene glycol, and formic and acetic mixed acid. Here, keeping the molar mass of the hydrogen bond donor the same, they were added to the flask. The degradation of lignin was carried out under nitrogen atmosphere with continuous stirring for 5 h at a temperature set at 150 °C under magnetic stirring.

After the reaction, the pH was adjusted to 2 by adding 1 mol/L HCl to the DES–lignin mixture, followed by high-speed centrifugation to obtain a partition of the regenerated lignin and the supernatant (DES–water mixture), and the pure regenerated lignin was obtained by repeated addition of large amounts of distilled water, and the supernatant was recovered. The recovered supernatant and regenerated lignin were washed thoroughly with ethyl acetate (extracted 3 times), and the regenerated lignin was freeze-dried for 24 h as a solid product. The ethyl acetate phase of the extracted supernatant was dried with anhydrous sodium sulfate and concentrated to remove the solvent as the liquid-phase product for further analysis (the organic phase was obtained as a phenolic-rich product after removing the solvent). DES was then recovered from the supernatant by rotary evaporation and dried for an additional 12 h at 105 °C. The DES degradation process and product separation is shown in Figure 1.

## 3. Results and Discussion

### 3.1. Determination of Phenolic Hydroxyl Groups

In the process of lignin degradation, the breakage of a large number of aryl ether bonds produces a large number of phenolic hydroxyl groups, and the content of phenolic hydroxyl groups in the degraded products is a key indicator when evaluating its degradation efficiency. The size of the phenolic hydroxyl group reflects the ability of DES to catalyze the cleavage of ether bonds [26]. In addition, the chemical reactivity of lignin in various modification processes is profoundly influenced by its phenolic hydroxyl content (e.g., reaction with formaldehyde to produce lignin phenolic resin binders [27]). Therefore, the quantification of phenolic hydroxyl groups can provide a relevant basis for understanding the structure and reactivity of lignin, as well as the mechanism and extent of lignin degradation.

The measurement of phenolic hydroxyl groups in the system was performed on partially degraded lignin using the Folin–Ciocalteu reagent method (FC method). It is based on the principle that the reaction of isophosphotungstate—molybdate with phenolic hydroxyl groups—appears blue and is strongly proportional to the number of total phenolic hydroxyl groups, and is suitable for spectrophotometric determination. This method allows a rapid and simple determination of phenolic hydroxyl groups in lignin [28].

Firstly, the standard curve was drawn. In this experiment, vanillin was used as the standard substance to draw the standard curve. The standard curve was drawn: 0.2013 g of vanillin was accurately weighed, dissolved in distilled water, and fixed in a 1000 mL volumetric flask. 10 mL of the solution was taken out from the volumetric flask, diluted to 100 mL with distilled water, and then 0, 1, 2, 4, 8, 10, 14 mL were drawn from it and placed in a 50 mL volumetric flask (the solution concentrations were 0, 4.278, 8.556, 17.112, 34.224, 42.780, 59.892 μmol/L). Then, add 3 mL of FC reagent and 30, 29, 28, 26, 22, 20, 16 mL of distilled water, shake well for 30 s, add 10 mL of 20% Na_2_C_2_O_4_ solution, mix well, add water to fix the volume, mix well again, and then stir the reaction for 2 h at room temperature to obtain a series of reaction products. The absorbance of the products was measured at 760 nm with a blank vanillin concentration of 0. The linear relationship between absorbance and vanillin concentration was obtained, as shown in Figure 1a. The contents of phenolic hydroxyl groups in different series with different temperatures were obtained, and the advantages and disadvantages of degradation in the system were conveniently observed by phenolic hydroxyl groups. A regression equation was developed based on the absorbance of vanillin at 760 nm versus concentration (µmol/L): *y* = 0.01391*x* + 0.0005 (correlation coefficient R^2^ = 0.9975).

The phenolic hydroxyl content of alkaline lignin obtained in this study was similar to that reported in the literature, ranging from 0.17 to 3.0 mmol/g [29,30]. Treatment of alkaline lignin with various types of DES increased the phenolic hydroxyl content of lignin and decreased the methoxy content of lignin [31]. As seen in Figure 1b, the degradation effect of acidic DES was generally higher than other types of DES, e.g., GE/CC, GC/CC, UR/CC. In addition, the highest concentration of phenolic hydroxyl groups was better degraded after TA/CC, indicating that strongly acidic DES contributed more to the degradation of lignin than weakly acidic DES. DES not only acted as a solvent, but also as a catalyst to promote the β-*O*-4′ bond cleavage at elevated reaction temperatures, leading to a sustained elevation of the phenolic OH group and a decrease in molecular weight [32].

### 3.2. FT-IR Characterization

In order to compare the structural changes of lignin before and after degradation, the FT-IR spectra were characterized for the alkaline lignin feedstock and the regenerated lignin obtained from different DES degradations (as shown in Figure 1c). It was found that the FT-IR spectra of degraded lignin and alkaline lignin were not much different, e.g., the absorption peaks near 1600 cm^−1^ and 1510 cm^−1^ representing the vibration of the aromatic skeleton of lignin did not change, indicating that the aromatic ring structure of lignin was not destroyed during the deep eutectic solvent treatment. The changes of the absorption peaks mainly occurred at 3305 cm^−1^ and 1078 cm^−1^ [33]. By studying the strength of the characteristic absorption peaks of each functional group of lignin, the qualitative data of each functional group could be obtained [23]. The vibration of -OH corresponding to the 3435 cm^−1^ peak was attenuated after degradation. It is presumed that this is due to the breakage of the ether bond in the lignin molecule during the deep eutectic solvent treatment to generate new hydroxyl groups. The regenerated lignin peaks of -OH and C-H were significantly weakened at 3435 cm^−1^ and 2941 cm^−1^, respectively, compared to the original lignin. It is presumed to be due to the breakage of the ether bond in the lignin molecule during the deep eutectic solvent treatment to generate new hydroxyl groups. The disappearance of 789 cm^−1^ and the appearance of 732 cm^−1^ corresponding to the aromatic ring vibrations suggest a change in the position of the phenyl substituent [34]. It is worth mentioning that the higher phenolic hydroxyl content resulting from degradation also contributes to the binding of DES to the regenerated lignin. This property differs from other lignin depolymerization methods. It is conceivable that the reduction or elimination of ether bonds in lignin may result in better polymer stability of regenerated lignin, since ether bonds are more susceptible to disruption than C-C bonds, offering the prospect of high-value applications of lignin [35].

### 3.3. GPC Characterization

To investigate the effect of deep eutectic solvent treatment on the molecular weight of lignin, the molecular weight of acetylated lignin before and after degradation was examined by GPC, as shown in Table 2. It can be seen that regenerated lignin M_w_ ranged from 2478 to 4330 g/mol, which was lower than the M_w_ value of the original alkaline lignin (7349 g/mol), the same as reported in the literature [36]. It is generally believed that the less the aryl ether bond content in lignin, the smaller the molecular weight. The less the condensed structure, the smaller the molecular weight. Combined with the FT-IR results, the DES-treated alkaline lignin had a lower ether bond content, while the molecular weight of the lignin was reduced. The M_w_ of TA/CC was higher than the other types, indicating that various reactions such as condensation occurred during the degradation process of this group of samples while depolymerizing and breaking the ether bonds; these findings can be found in the study of Li, et al. [23]. In addition, it has also been shown that the intramolecular hydrogen bonds and carbonyl groups of lignin are detrimental to the carboxylic acid-induced cleavage of the C-O bonds of lignin molecules [37]. The polydispersity coefficient (M_w_/M_n_) of lignin was significantly lower after degradation, indicating that the degree of homogeneity of lignin was improved after degradation. The molecular weights of TACC, OX/CC, AA/CC, MA/CC, GE/CC, GC/CC, and UR/CC were opposite to the pattern of phenolic hydroxyl groups in the supernatant mentioned above, presumably because more ether bonds were broken under acidic conditions and easily polymerized into polymorphs. The ether bonds were partially broken under other conditions, and the short-chain structure of lignin remained. Acid DES-treated lignin has low molecular weight and high reactivity, while basic DES lignin maintains the main natural structure but with a lower molecular weight. The homogeneous dispersion and low molecular weight promote the downstream conversion and utilization of lignin macromolecules [38].

### 3.4. TG Characterization

Figure 2a shows the thermogravimetric analysis of alkaline lignin and regenerated lignin. As can be seen from the figure, the initial phase is the drying phase of the lignin sample (30–150 °C). The first peak of weight loss after 150 °C corresponds to the individual interaction reaction between the phenolic hydroxyl epoxy groups in lignin, mainly involving the degradation of β-*O*-4; these results are consistent with Poletto’s study [39]. During the initial pyrolysis phase (200–350 °C), the destruction of low-molecular-weight lignin fragments and the breakage of side chains in the lignin fraction occur [40]. During the subsequent degradation at 350–400 °C, the degradation rate of lignin mainly depends on the side-chain oxidation of lignin, such as carbonylation, carboxylation, and dehydrogenation reactions. At temperatures greater than 400 °C, the rate of pyrolysis gradually slows down, indicating a relatively stable aromatic ring structure and slow weight loss, with a series of reactions occurring mainly on the aromatic ring of lignin, such as saturation of the aromatic ring, carbon–carbon bond breakage, and degradation of lignin to CO_2_, CO, and H_2_O [41]. At a given endpoint of testing at 700 °C, the weight loss was 71.45% for TA/CC, 49.09% for alkaline lignin, and ranged from 41.71% to 46.51% for GE/CC, OX/CC, GC/CC, MA/CC, UR/CC, and AA/CC. It can be found that the thermal stability of most of the regenerated lignin is lower than that of alkaline lignin, while the thermal stability of TA/CC is much better than that of alkaline lignin. The relatively large weight loss rates of GE/CC, OX/CC, GC/CC, MA/CC at 200–600 °C indicated that the β-*O*-4 linkages and lignin side chains were still retained after the treatment process. The samples of GE/CC, OX/CC, GC/CC, MA/CC reached the fastest pyrolysis rate at the β-O-4 pyrolysis stage, while UR/CC and AA/CC reached the fastest pyrolysis rate in the side-chain pyrolysis phase, indicating more side-chain structures [42]. In a variety of DES systems, the DES of choline chloride/oxalic acid broke the structure of part of the side-chain region of lignin, and the lignin molecules with broken side chains re-formed intra- or intermolecular C-C and C-O bonds, causing the lignin to condense [43]. It has been reported in the literature that methanol promotes the esterification reaction of carboxyl groups at high temperatures and acidic conditions and inhibits the repolymerization of lignin with free radicals [44]. It can be seen that the alcohol DES contributes to the reduction in the condensation reaction activity of regenerated lignin, which is consistent with the results of the thermogravimetric curves.

### 3.5. ^1^H NMR Characterization

Figure 2b shows the ^1^H-NMR hydrogen spectra of alkaline lignin and regenerated lignin degraded by different DES species. Among them, 8.0–6.2 ppm are attributed to the proton signals on the aromatic nuclei of guaiacyl unit and lilacyl unit, 5.0–4.1 ppm are attributed to the H_α_, H_β_, and H_γ_ signals in the β-*O*-4 structure, and 4.1–3.5 ppm are attributed to the methoxy proton signal. The sharp peak at 3.5–3.3 ppm is attributed to the proton signal of water in the solvent, and the small peak at 3.4 ppm is attributed to the proton signal of choline chloride, which may be due to a slight reaction between lignin and choline chloride in the solvent or a small amount of deep eutectic solvent remaining in the lignin during the treatment. 2.5–2.2 ppm is attributed to the proton signal of aromatic ring acetate, and 2.2–1.4 ppm is attributed to the proton signal of aliphatic acetate [23]. The hydrogen of the phenolic hydroxyl group is very active, and the solvent is taken off in the deuterated reagent. In general, aromatic phenolic hydroxyl groups tend to be more low-field. The alcohol hydroxyl group is 0.5–5.5 ppm and the phenolic hydroxyl group is 4–8 ppm. In addition, the phenolic hydroxyl group may also be seen at chemical shifts greater than eight. The breakage of the β-*O*-4 bond generates new phenolic hydroxyl functional groups, while the breakage of the C_α_-C_β_ bond increases the alcohol hydroxyl content in lignin. After regenerating lignin with *p*-toluenesulfonic acid as the hydrogen bond donor, more carbon–carbon bonds were broken, while AA/CC and MA/CC were also partially broken; it can be seen that acidic deep eutectic solvents are more effective for lignin degradation. The methoxy content of the regenerated lignin after acidic DES treatment was slightly higher than that of AL, as seen by ^1^H-NMR. This could be due to the further reaction between lignin and acid during the extraction of lignin to produce lignin derivatives (Lignin-O-CR) [45]. Based on the results of the present study, significant lignin depolymerization and condensation can be observed in acidic DES lignin [8]. It is worth mentioning that the complex and polymeric structure of lignin is responsible for the overlapping signals of ^1^H NMR spectra [46].

### 3.6. Recyclability of Solvents

The recyclability of solvents can reduce environmental pollution and save reaction costs. We recovered seven DESs using rotary evaporation and discussed the recovery rates of DESs accordingly. As shown in Figure 2c, the DES recoveries were above 90% for GE/CC, UR/CC, GC/CC, and MA/CC, and above 80% for TA/CC, OX/CC, and AA/CC. In particular, the DES recoveries of MA/CC and MA/CC were slightly lower than those of GE/CC, UR/CC, and GC/CC due to the volatilization of formic acid and acetic acid during the actual reaction and recovery. The advantages of DES over conventional solvents are its low toxicity, non-volatility, low cost, and recyclability. In addition, the viscosity of DES, which typically ranges from 10 to 5000 cP, is strongly influenced by temperature. For example, as the temperature increases from 30 °C to 95 °C, the viscosity of DES for UR/CC decreases sharply from 500 cP to 50 cP. It has been reported that the high viscosity of DES may lead to difficulties in reuse [47]. This can be understood by the formation of DES between HBA (ChCl) and HBD (GE, UR, GC, MA, TA, OX, and AA) by a strong hydrogen bonding network [48]. Moreover, the loss of solvent attached to the solute, the water content of DES after recovery was reduced, which could be the reason for the decrease in DES recovery.

## 4. Conclusions

In this study, we propose a green solvent system for lignin degradation with low toxicity and reusability of DES compared to other solvents. This study proposes a good solvent system for the efficient and high-quality degradation of lignin. Lignin is a complex macromolecular structure with complicated bond types, and the effect of each type of DES on lignin degradation is different, so it is also critical to find a good degradation system. Seven different DESs were used to degrade alkaline lignin under given conditions and the supernatant and regenerated lignin after their degradation were analyzed, and it was found that the phenolic hydroxyl content in the supernatant increased a lot (126–481% compared to alkaline lignin). The regenerated lignin fraction showed attractive structural and functional characteristics (homogeneous dispersion with low molecular weight). Acid DES-treated lignin exhibited higher fragmentation (i.e., cleavage of β-*O*-4 bonds) and condensed structures and lower molecular weights, and TA/CC DES was more degraded and had the highest degree of condensation in acid DES. The thermal stability of most of the regenerated lignin was lower than that of AL. The breakage of the β-*O*-4 aryl ether bond in lignin generates phenolic hydroxyl functional groups, while the breakage of the C-C bond increases the content of alcohol hydroxyl groups in degraded lignin. In addition, the performance applications of the regenerated lignin are further explored, and the degraded regenerated lignin can be applied to phenolic resins, hydrogels, and carbon aerogels.

## Data Availability

Data sharing is not applicable to this article as no datasets were generated or analyzed during the current study.

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
