# Peer review of "Comparison of the Degradation Performance of Seven Different Choline Chloride-Based DES Systems on Alkaline Lignin"

_polymers, 2022, doi:10.3390/polym14235100_

Round 1

Reviewer 1 Report

The manuscript entitled "Comparison of the degradation performance of different choline chloride-based DES systems on lignin polymers" reported an excellent green degradation method for degradation of lignin by heating with deep eutectic solvent. GPC observed the molecular weight of lignin after degradation and found a significant increase in the homogeneity (1.6-2.0) and a significant decrease in the molecular weight Mw (2478-4330) of the regenerated lignin. Seven DES solutions were recovered and it was found that the recovery of DES still reached more than 80% at the first recovery. In general, it is an interesting and valuable topic to deserving a research article.

However, there are still some problems to be solved. So this reviewer would suggest a moderate revision before its acceptance.

1.      It is better to give the full name in the title.

2.      Scheme 1 is too cluttered as a flowchart and needs to be simplified, especially for the text content on the diagram. Please submit the reworked Scheme 1 in the new manuscript.

3.      Figure 3 needs to be reformatted.

4.      Equations must be typeset using appropriate fonts and layout. Excel figures must not contain the superfluous outer frame. The same information must not be duplicated in Tables and Figures.

5.      More introduction on the DES and lignin should be provided with supporting articles: LI Lifen,WU Zhigang,LIANG Jiankun,YU Liping*.Application of deep eutectic solvents in lignocellulosic biomass processing[J].Journal of Forestry Engineering,2020,5(04):20-28.doi:10.13360/j.issn.2096-1359.201907035; YU Yanyan,LI Yilin,LOU Yuhan,LIU Yongzhuang,YU Haipeng.Effect of lignin condensation on cellulose enzymatic hydrolysis during deep eutectic solvent fractionation of lignocellulose[J].Journal of Forestry Engineering,2021,6(06):101-108.doi:10.13360/j.issn.2096-1359.202104022; etc.

6.      The format of the caption is not uniform.

7.      Table 1 should be provided in a more scientific way with three-line table.

8.      It is recommended not to put a lot of single images in the article, and there should be more content in these tests that can be shown through images. Please submit the revised pictures in the manuscript again.

9.      Although authors have proposed many problems and challenges, possible solutions to solve these challenges is better to be also suggested.

10.  The mechanism of the effect of DES on the lignin should be further explained. Please refer: WANG Lei,LOU Yuhan,TONG Zhihan,MENG Juan,SHI Xiaochao,CAO Kaiyue,XIA Qinqin,YU Haipeng.Molecular dynamics mechanism of metal salt hydrate-based deep eutectic solvent to dissolve cellulose at room temperature[J].Journal of Forestry Engineering,2022,7(04):64-71.doi:10.13360/ j.issn.2096-1359.202112025

11.  Please carefully check the whole manuscript. There are still some typos and grammar issues. In addition, please carefully check the references to ensure the full information is included.

Author Response

We thank the reviewers for their careful review of our manuscript, which we have adequately revised.
1. We have renamed the title of the manuscript.

2. We have redrawn Scenario 1 and simplified the roadmap.

3. Figure 3 has been reformatted.

4. we have reworked the information in the tables and figures.

5. We have added two literature recommended by the reviewers to support the article. For example, L.F. Li,C.G. Wu,J.K. Liang, et al. Journal of Forestry Engineering,2020,5(04):20-28.DOI:10.13360/j.issn.2096-1359 .201907035;Yu Yanyan,Li Yilin,Lou Yuhan,et al. Journal of Forestry Engineering,2021,6(06):101-108.DOI:10.13360/j.issn.2096-1359.202104022.

6. The format of the title has been reformatted.

7. The table has been changed to a three-line table.

8. It is recommended not to put a large number of single images in the article, there should be more content in these experiments that can be shown by images. Please submit revised images again in the manuscript.

9. The conclusions of the article have been redescribed.

10. We have added to the results and discussion in the article to further explain the mechanism of DES effect on lignin.

11. We double-checked the entire manuscript with references.

Reviewer 2 Report

Overall, based on the research and the data, it is appropriate for publication in this journal. The article as a whole is fine. However, there are some issues that have to be fixed before publication;

Abstract should be revised for better explaining the paper content.

In introduction, mention some other related studies of the concerned study.

How the author compare with this series, is the amount taken was same for all hydrogen bond donor?

Need to improve the language of manuscript and also recheck the full article as there are so many mistakes of typo/formatting/uniformity etc.

Need to add more reference in result and discussion section to support the results.

Need to improve the quality of figures.

Make sure that the format of references are uniform.

Conclusion also revised based on the results.

Some errors regarding the sub/super script, spacing and typo need to consider throughout the manuscript.

Author Response

We thank the reviewers for their comments and suggestions on our manuscript, which has been fully revised.
1. We have made additional revisions to the abstract to better explain the content of the paper.

2. We added many references in the introduction.

3. We added the key information that all hydrogen bond donors have the same uptake.

4. We improved the language of the manuscript and rechecked the whole text.

5. We added 6 references in the Results and Discussion section to support the results.

6. We redrew the images in the manuscript.

7. We determined the format of the citations.

8. We revised the conclusions based on the results.

Round 2

Reviewer 1 Report

Accept in present form

Author Response

Thanks to the reviewers for their positive comments.